# Patterns of motivators and barriers to heart health behaviors among adults with behavior-modifiable cardiovascular risk factors: A population-based survey in Singapore

Zijuan Huang[1‡], Jien Sze Ho[1‡], Qai Ven Yap[2], Yiong Huak Chan[2], Swee Yaw Tan[1], Natalie Koh Si Ya[1], Lip Ping Low[3], Huay Cheem Tan[4], Woon Puay Koh[5,6], Terrance Siang Jin Chua[1], Sungwon Yoon[7,8]*

1 National Heart Centre Singapore, Singapore, Singapore, 2 Department of Biostatistics, Yong Loo Lin School of Medicine, Singapore, Singapore, 3 Low Cardiology Clinic, Mount Elizabeth Medical Centre, Singapore, Singapore, 4 Department of Cardiology, National University Heart Centre, Singapore, Singapore, 5 Healthy Longevity Translational Research Programme, Yong Loo Lin School of Medicine, National University of Singapore, Singapore, Singapore, 6 Singapore Institute for Clinical Sciences, Agency for Science Technology and Research (A*STAR), Singapore, Singapore, 7 Health Services and Systems Research, Duke-NUS Medical School, Singapore, Singapore, 8 Centre for Population Health Research and Implementation (CPHRI), SingHealth Regional Health System, SingHealth, Singapore, Singapore

‡ ZH and JSH are co-first authors and contributed equally to this work.
* Sungwon.yoon@duke-nus.edu.sg

**Data Availability Statement:** All relevant data are within the manuscript and its Supporting information files.

## Abstract

### Objectives

Motivators and barriers are pivotal factors in the adoption of health behaviors. This study aims to identify patterns of the motivators and barriers influencing heart health behaviors among multi-ethnic Asian adults with behavior-modifiable risk factors for heart disease, namely obesity, physical inactivity and smoking.

### Methods

A population-based survey of 1,000 participants was conducted in Singapore. Participants were assessed for behavior-modifiable risk factors and asked about motivators and barriers to heart health behaviors. Exploratory and confirmatory factor analyses were conducted to identify factors underlying motivator and barrier question items. Logistic regression was conducted to examine the associations of motivator and barrier factors with sociodemographic characteristics.

### Results

The twenty-five motivator and barrier items were classified into three (outcome expectations, external cues and significant others including family and friends) and four (external circumstances, limited self-efficacy and competence, lack of perceived susceptibility, benefits and intentions and perceived lack of physical capability) factors respectively. Among

**Funding:** This work was supported by funding from the Singapore Heart Foundation. The funders had no role in study design, data collection and analysis, decision to publish, or preparation of the manuscript.

**Competing interests:** The authors have declared that no competing interests exist.

participants with behavior-modifiable risk factors, those with lower education were more likely to be low in motivation factor of "outcome expectations" and "external cues". The well-educated were more likely to be high in the barrier factor of "lack of perceived susceptibility, benefits and intention" and were less likely to have the motivation factor of "significant others (family or friends)". Those aged 60–75 years had low motivations and high barriers compared to their younger counterparts. Older age was more likely to be low in motivation factor of "outcome expectations" and "external cues" and high in barrier factor of "limited self-efficacy and competence" and "perceived lack of physical capability".

## Conclusions

Findings underscore the importance of a targeted intervention and communication strategy addressing specific motivation and barrier factors in different population segments with modifiable risk factors.

## Introduction

Cardiovascular disease (CVD) is largely preventable by addressing modifiable risk factors and adopting heart health behaviors [1–3]. Despite positive health benefits of heart health behaviors, a significant proportion of adults across different populations still lack adequate heart healthy behaviors [4–6]. It is also increasingly evident that knowledge alone is insufficient to change behavior with multiple studies demonstrating a significant disconnect between knowledge and behavior [7, 8].

Evidence demonstrates that ability or inability to initiate and sustain heart health behaviors can be explained by motivators and barriers [9, 10]. Without understanding factors that enable or impede an individual's ability to change health behaviors, heart disease prevention efforts may be unlikely to achieve their intended impact. This is particularly important for those whose CVD risk factors are modifiable, but do not undertake healthy behaviors.

Existing literature suggests that various factors influence the engagement in heart healthy behaviors among adults. They include support from family and friends, beliefs about the causes of illness, lack of time and access issues such as transport and financial costs [11–13]. However, few studies have examined Asian adults at the population level. In addition, existing population surveys tended to focus on population in general as opposed to a population segment who have modifiable CVD risks through behavior changes. As populations may differ in social, cultural and environmental aspects, it is important to conduct population-specific and segment-based assessment of factors that influence heart healthy behaviors. Since the motivators and barriers are often distinct and heterogenous across individuals, categorizing them into several broad components would allow for understanding the patterns of motivators and barriers thereby guiding future efforts for health promotion.

To this end, we conducted a population survey to examine the factors that motivate and hinder uptake of heart health behavior in our Asian population. These motivators and barriers have previously been surveyed in the Western population [14]. The present study examined patterns of motivators and barriers among adult segments who have behavior-modifiable risk factors for heart disease—obesity [15], physical inactivity [16] and smoking [17, 18]. We also sought to examine the association of the patterns of motivators and barriers and sociodemographic characteristics among these three different risk groups.

## Methods

### Sample

This is a population-based survey conducted in Singapore. Eligibility criteria included Singapore citizens or permanent residents of both genders aged 21 to 75 years old. Stratified cluster random sampling was used. A list of addresses was obtained from the Singapore Department of Statistics. The residential areas were divided into 26 subzones, and selection was stratified by subzone then by housing type (public housing, landed property, condominium, shop house). Within each housing type, blocks and units were randomly selected. There was a quota for gender and age group to ensure that participants' demographics were matched with the national distribution. We oversampled certain ethnic groups to obtain a nearly equal proportion of three major ethnic groups in Singapore (i.e. Chinese, Malays and Indians). The race distribution in Singapore is approximately 70% Chinese, 20% Malay and 10% Indian.

### Instrument

We used a structured questionnaire to collect information on socio-demographics, behavior-modifiable risk factors and motivators and barriers to the uptake of heart healthy behaviors.

**Sociodemographic variables.** We collected age, gender, ethnicity, education and employment status. Education was classified into "primary or below" (primary school or lower level of education), "university or above" (university level education or above) and "intermediate" (education level between "primary or below" and "university or above"). Employment status was classified into "working"' (full time employed, part time employed, self-employed, employers), "retired/homemakers" (retirees and homemakers), "others" (national servicemen, students, unemployed).

**Behavior-modifiable cardiovascular risk factors: Physical inactivity, smoking and obesity.** Participants were asked about their physical activities in the past month with a series of questions adopted from the Behavioral Risk Factor Surveillance System (BRFSS) [19]. Physical activities cited were matched to their respective metabolic equivalent of task [20] (MET) values based on the BRFSS activity table and classified into low, moderate and vigorous intensity exercise. Time spent in activities per week was computed based on the answers given on time spent on each activity. The MET data was then dichotomized into "adequate exercise" (which met heart healthy activity recommendations of 150 min per week of moderate intensity aerobic exercise or 75 min per week of vigorous aerobic exercise or equivalent combinations) and "inadequate exercise" (which did not meet heart healthy activity recommendations). Physical inactivity was defined as not achieving the recommendations (i.e., inadequate exercise). Participants were also asked about smoking status [21], whether they had smoked 100 cigarettes in their lifetime with options being 'yes', 'no', 'don't know'. If they answered 'yes', they were classified as "ever smokers". Ever smokers were then asked whether they currently were smoking 'every day', 'some days' or 'not at all'. 'Not at all' response was classified as "not currently smoking". 'Some days' and 'every day' responses were classified as "current smokers". Lastly, participants' self-reported height (cm) and weight (kg) were collected, body mass index [22] (BMI) was then calculated from these values. BMI categories were defined using national cut off points, that is BMI $\geq 27.5$ for obese, $23 \leq BMI < 27.5$ for overweight, BMI $< 23$ for normal or low BMI.

**Motivators and barriers.** Items on motivators and barriers to lifestyle changes were developed based on a similar study conducted in the United States [14]. The items on motivators and barriers to lifestyle changes presented a high internal consistency, both with Cronbach's alpha 0.904. Participants were presented with examples of heart healthy actions (e.g., quit smoking, have regular physical exercise, lose weight, reduce cholesterol intake, reduce stress,

reduce sodium intake, maintain a healthy blood pressure, moderate alcohol consumption, increase fruits/vegetable intake, get adequate sleep, reduce sugar intake, visit a doctor to get regular tests for heart disease prevention). Participants were asked if they had taken some of these actions, how much was driven by certain motivators listed (e.g. because I live longer, I was encouraged by family members); and if they had not taken these actions, how much was due to certain barriers listed (e.g. I don't think I am at risk of heart disease, I have no time to change my lifestyle) on a 4-point Likert scale of agreement (1 = strongly agree/2 = agree/3 = disagree/4 = strongly disagree). Each response to motivator items was reversely coded. Total score was calculated and dichotomized into low motivation (<mean score-1SD) and high motivation (>= mean score-1SD). Each response to barrier items was summed and dichotomized into high barrier (<mean score-1SD) and low barrier (>= mean score-1SD).

## Procedure

The questionnaire was translated into English, Chinese, Malay and Tamil. A forwards-backwards translation was performed to ensure that the questionnaires were conceptually equivalent. An initial pilot survey was tested in 10 respondents and revisions were made based on feedback provided. Trained interviewers went door-to-door to administer the survey. The survey was administered in one of the four national languages (English, Mandarin, Malay or Tamil), and was carried out from April 2018 to May 2018. As the survey was anonymous, verbal consent was obtained by the interviewer before the survey was conducted. This study was approved by National University of Singapore's Institutional Review Board (S-17-256E).

## Sample size

From prior estimates of individuals with suboptimal heart health behavior such as lack of exercise in Singapore, about one third did not have adequate exercise [23]. Planning for a 1-month recruitment period, 1,000 participants were invited. This would provide a precision of 3% on a postulated prevalence of 30% (highest) for any one of the three behavior-modifiable risk factors to be studied.

## Statistical analysis

Descriptive statistics for socio-demographic numerical variables were presented as mean (SD) and n (%) for categorical variables. Cronbach alphas were presented to show the reliability of the question items on motivators and barriers. Factor analysis was conducted to classify motivator and barrier question items into groups. This was done by first conducting exploratory factor analysis, to assess the underlying factor structure of motivator and barrier question items. This was validated by confirmatory factor analysis (CFA). The factors that the motivators and barrier questions were grouped under were then defined. The associations of motivators, barriers and socio-demographics in adult population segments with behavior-modifiable risk factors (physical inactivity, smoking, obesity) were determined using weighted (by race) logistic regression. Variables with p-value<0.2 from univariate logistic regression were entered into multivariate logistic regression models. Statistical analysis was performed using STATA version 16 with statistical significance set at 2-sided $p < 0.05$.

## Results

### Sociodemographic characteristics

Of 1,000 participants who were interviewed, 357(35.7%), 321(32.1%), 322(32.2%) were Chinese, Malays and Indian, respectively. All were Singaporeans or permanent residents in

**Table 1. Demographics of survey respondents (n = 1,000).**

| Variable | Category | Whole Cohort | Chinese | Malay | Indian |
|---|---|---|---|---|---|
| Age group | 20–39 years old | 385 | 123 (34.5) | 138 (43) | 124 (38.5) |
| | 40–59 years old | 395 | 137 (38.4) | 124 (38.6) | 134 (41.6) |
| | 60–75 years old | 220 | 97 (27.2) | 59 (18.4) | 64 (19.9) |
| Gender | Female | 521 | 186 (52.1) | 168 (52.3) | 167 (51.9) |
| | Male | 479 | 171 (47.9) | 153 (47.7) | 155 (48.1) |
| Education level | Low education (primary school or less) | 194 | 65 (18.2) | 70 (21.8) | 59 (18.3) |
| | Intermediate education | 679 | 216 (60.5) | 234 (72.9) | 229 (71.1) |
| | High education (university or above) | 127 | 76 (21.3) | 17 (5.3) | 34 (10.6) |
| Employment | Employed full time | 551 | 197 (35.75) | 177 (32.12) | 177 (32.12) |
| | Employed part time | 78 | 22 (28.21) | 28 (35.90) | 28 (35.90) |
| | Employer/self employed | 36 | 19(52.78) | 8(22.22) | 9(25.00) |
| | Unemployed looking for work | 13 | 1(7.69) | 7(53.85) | 5(38.46) |
| | Not working not looking for work | 39 | 11(28.21) | 10(25.64) | 18(46.15) |
| | Retired | 94 | 42(44.68) | 27(28.72) | 25(26.60) |
| | Homemaker | 141 | 49(34.75) | 45(31.91) | 47(33.33) |
| | Full time National Service | 10 | 4(40.00) | 6(60.00) | 0(0) |
| | Full time student | 38 | 12 (31.58) | 13 (34.21) | 13 (34.21) |
| BMI | Low risk | 418 | 173 (48.46) | 131 (40.81) | 114 (35.4) |
| | Moderate risk | 417 | 143 (40.06) | 126 (39.25) | 148 (45.96) |
| | High risk | 165 | 41 (11.48) | 64 (19.94) | 60 (18.63) |

Note: Values are in n (%).

Singapore. More than two thirds of participants had an intermediate level of educational attainment and about two thirds were in the workforce (Table 1). There was no missing data.

## Exploratory factor analysis (EFA)

EFA classified motivator question items into three factors and barrier question items into four factors. The items were assigned to the factor on which it had the highest factor loading. The three factors of motivator question items were named *outcome expectations*, *external cues* and *significant others (family or friends)*. While the four factors of barrier question items were named e*xternal circumstances*, *limited self-efficacy and competence*, *lack of perceived susceptibility*, *benefits and intentions* and *perceived lack of physical capability* (Table 2).

## Confirmatory factor analysis (CFA)

CFA confirmed the factor structure identified by EFA with high goodness of fit. (All CFI>0.9) (Table 2).

## The physically inactive

**Total scores of motivators and barriers in the physically inactive.** Among physically inactive individuals, lower education (primary level education or below) was significantly associated with having low motivations (adjusted OR 2.4, 95% CI 1.4–4.2, p = 0.001) as compared to intermediate level education. Those with intermediate level education and university level education or above were significantly more likely to have high barriers as compared to those with primary level education or below (adjusted OR 2.4, 95% CI 1.1–4.9, p = 0.022 and adjusted OR 5.1, 95% CI 2.1–12.5, p<0.001, respectively) (Table 3).

**Table 2. Exploratory and confirmatory factor analysis.**

| Factors | Items | Factor loadings | CFI |
|---|---|---|---|
| **Motivators** | | | |
| Outcome expectations | Live longer | 0.727 | 0.935 |
| | Feel better | 0.830 | |
| | Improve my health | 0.806 | |
| | Avoid taking medications | 0.784 | |
| | Do it for my family | 0.759 | |
| External cues | Doctor/nurse | 0.594 | 0.989 |
| | Information on heart disease in the media (newspapers and TV) | 0.857 | |
| | Community health events | 0.882 | |
| | Death and illness of family members from heart disease | 0.731 | |
| Significant others (family or friends) | Family members | 0.800 | 1.000 |
| | Friends | 0.792 | |
| **Barriers** | | | |
| External circumstances | I have too many things on my mind/distracted/depressed/preoccupied/stressed to change my lifestyle. | 0.525 | 0.975 |
| | I don't have the money to change my lifestyle. | 0.757 | |
| | I have no time to change my lifestyle. | 0.800 | |
| | I feel the changes required are too difficult. | 0.769 | |
| | I don't know what I should do. | 0.669 | |
| | There is too much confusion in the media about what to do to change my lifestyle. | 0.488 | |
| Limited self-efficacy and competence | I have tried but failed to change my lifestyle. | 0.768 | 1.000 |
| | I'm not confident that I can change my lifestyle. | 0.661 | |
| | My doctor doesn't explain clearly what I should do. | 0.736 | |
| Lack of perceived susceptibility, benefits and intentions | I don't think I am at risk of heart disease. | 0.851 | 1.000 |
| | I don't want to change my lifestyle. | 0.856 | |
| | Changing my behavior will not reduce my risk of heart disease. | 0.732 | |
| Perceived lack of physical capability | I am too ill to change my lifestyle. | 0.756 | 1.000 |
| | I am too old to change my lifestyle. | 0.876 | |

**Predictors of motivator factors in the physically inactive.** Significant differences for some motivator factors were noted by sociodemographic characteristics and BMI. Intermediate level education was significantly associated with lower motivation as compared to university level education or above (adjusted OR 1.9, 95% CI 1.1–3.2, p = 0.018). For the motivator factor "external cues", primary education or below was significantly associated with being low in this factor compared to those of intermediate education (adjusted OR 2.0, 95% CI 1.2–3.2, p = 0.004). Those low in the motivator factor of "significant others (family or friends)" were significantly more likely to be those with university level education or above compared to those with intermediate education (adjusted OR 4.5, 95% CI 1.9–10.7, p = 0.001). For "outcome expectations", those aged 60–75 seemed more likely to have low motivations as compared to those aged 40–59, with borderline significance (adjusted OR 1.569, 95% CI 1.004–2.453, p = 0.048) (Table 3).

**Predictors of barrier factors in the physically inactive.** Significant differences for some barrier factors were noted by sociodemographic characteristics and BMI. For the "limited self-efficacy and competence" factor, those with university level education or above were significantly more likely to be high in this barrier factor as compared to primary level education or below (adjusted OR 3.5, 95% CI 1.5–8.4, p = 0.004). For the "lack of perceived susceptibility,

**Table 3. Motivators and barriers among the physically inactive: Overall score and factors.**

| Motivators (total score) | | | | | | |
|---|---|---|---|---|---|---|
| **Variable** | **Motivated** | **Very Unmotivated** | **Unadjusted** | | **Adjusted** | |
| | | | **OR (95% CI)** | **P-value** | **OR (95% CI)** | **P- value** |
| **Race** | | | | | | |
| Chinese | 361(80.0%) | 90(20.0%) | 1.6(0.9–2.8) | 0.091 | 1.7(1.0–3.0) | 0.070 |
| Indian | 56(80.0%) | 14(20.0%) | 1.6(0.7–3.4) | | 1.6(0.7–3.7) | |
| Malay | 116(86.6%) | 18(13.4%) | 1.0 | 0.253 | 1.0 | 0.217 |
| **Age group** | | | | | | |
| 20–39 | 183(81.0%) | 43(19.0%) | 1.2(0.8–2.0) | 0.388 | 1.5(0.9–2.4) | 0.114 |
| 60–75 | 121(77.1%) | 36(22.9%) | 1.6(1.0–2.6) | | 1.2(0.6–2.2) | |
| 40–59 | 228(84.1%) | 43(15.9%) | 1.0 | 0.068 | 1.0 | 0.574 |
| **Education** | | | | | | |
| Primary or below | 106(71.6%) | 42(28.4%) | 2.3(1.4–3.6) | <0.001 | 2.4(1.4–4.2) | 0.001 |
| University or above | 85(80.2%) | 21(19.8%) | 1.4(0.8–2.4) | 0.256 | 1.4(0.8–2.6) | 0.213 |
| Intermediate | 342(85.1%) | 60(14.9%) | 1.0 | | 1.0 | |
| **Employment status** | | | | | | |
| Others | 39(70.9%) | 16(29.1%) | 2.1(1.1–4.0) | 0.018 | 2.1(1.1–4.1) | 0.035 |
| Retired/homemakers | 116(79.5%) | 30(20.5%) | 1.3(0.8–2.1) | 0.275 | 1.0(0.5–1.9) | 0.961 |
| Working | 378(83.4%) | 75(16.6%) | 1.0 | | 1.0 | |
| Motivator factors | | | | | | |
| **Variable** | **High score** | **Very low score** | **Unadjusted** | | **Adjusted** | |
| | | | **OR (95% CI)** | **P-value** | **OR (95% CI)** | **P- value** |
| Outcome expectations | | | | | | |
| **Age group** | | | | | | |
| 20–39 | 156(69.0%) | 70(31.0%) | 1.2(0.8–1.7) | 0.424 | 1.2(0.8–1.8) | 0.361 |
| 60–75 | 97(61.8%) | 60(38.2%) | 1.6(1.1–2.5) | 0.023 | 1.569(1.004–2.453) | 0.048 |
| 40–59 | 196(72.3%) | 75(27.7%) | 1.0 | | 1.0 | |
| **Education** | | | | | | |
| Primary or below | 99(67.3%) | 48(32.7%) | 1.9(1.1–3.5) | 0.028 | 1.7(0.9–3.2) | 0.129 |
| Intermediate | 266(66.3%) | 135(33.7%) | 2.0(1.2–3.4) | 0.007 | 1.9(1.1–3.2) | 0.018 |
| University or above | 85(80.2%) | 21(19.8%) | 1.0 | | 1.0 | |
| **Gender** | | | | | | |
| Male | 219(66.0%) | 113(34.0%) | 1.3(0.9–1.8) | 0.128 | 1.2(0.9–1.7) | 0.279 |
| Female | 231(71.5%) | 92(28.5%) | 1.0 | | 1.0 | |
| **Currently smoking** | | | | | | |
| Yes | 106(62.4%) | 64(37.6%) | 1.5(1.0–2.1) | 0.045 | 1.3(0.9–2.0) | 0.132 |
| No | 344(70.9%) | 141(29.1%) | 1.0 | | 1.0 | |
| External cues | | | | | | |
| **Education** | | | | | | |
| Primary or below | 98(66.2%) | 50(33.8%) | 1.9(1.2–2.8) | 0.004 | 2.0(1.2–3.2) | 0.004 |
| University or above | 75(70.8%) | 31(29.2%) | 1.5(0.9–2.4) | 0.094 | 1.6(1.0–2.6) | 0.057 |
| Intermediate | 315(78.6%) | 86(21.4%) | 1.0 | | 1.0 | |
| **Employment status** | | | | | | |
| Working | 343(75.7%) | 110(24.3%) | 0.97(0.63–1.49) | 0.871 | 1.2(0.7–2.0) | 0.517 |
| Others | 34(61.8%) | 21(38.2%) | 1.8(0.9–3.5) | 0.080 | 2.2(1.1–4.4) | 0.028 |
| Retired/homemakers | 110(75.3%) | 36(24.7%) | 1.0 | | 1.0 | |
| **Currently smoking** | | | | | | |
| Yes | 115(67.6%) | 55(32.4%) | 1.6(1.1–2.4) | 0.015 | 1.6(1.1–2.4) | 0.019 |
| No | 373(77.1%) | 111(22.9%) | 1.0 | | 1.0 | |

(*Continued*)

**Table 3.** (Continued)

| | Significant others (family or friends) | | | | | |
|---|---|---|---|---|---|---|
| **BMI** | | | | | | |
| Overweight | 262(93.6%) | 18(6.4%) | 1.7(0.8–3.6) | 0.202 | 1.7(0.8–3.8) | 0.173 |
| Obese | 92(91.1%) | 9(8.9%) | 2.3(0.9–5.8) | 0.077 | 2.5(0.9–6.6) | 0.066 |
| Low or normal | 263(96.0%) | 11(4.0%) | 1.0 | | 1.0 | |
| **Education** | | | | | | |
| Primary or below | 136(92.5%) | 11(7.5%) | 2.3(1.0–5.1) | 0.043 | 2.1(0.9–5.3) | 0.095 |
| University or above | 94(88.7%) | 12(11.3%) | 3.4(1.5–7.7) | 0.003 | 4.5(1.9–10.7) | 0.001 |
| Intermediate | 387(96.5%) | 14(3.5%) | 1.0 | | 1.0 | |
| **Employment status** | | | | | | |
| Others | 47(85.5%) | 8(14.5%) | 3.4(1.4–8.1) | 0.005 | 4.3(1.7–11.0) | 0.002 |
| Retired/homemakers | 139(95.2%) | 7(4.8%) | 1.0(0.5–2.5) | 0.887 | 1.0(0.4–2.8) | 0.973 |
| Working | 431(95.1%) | 22(4.9%) | 1.0 | | 1.0 | |

| | Barriers (total score) | | | | | |
|---|---|---|---|---|---|---|
| **Variable** | **Low** | **Very high** | **Unadjusted** | | **Adjusted** | |
| | | | **OR (95% CI)** | **P-value** | **OR (95% CI)** | **P- value** |
| **BMI** | | | | | | |
| Overweight | 250(89.3%) | 30(10.7%) | 1.1(0.6–1.9) | 0.775 | 1.1(0.6–2.0) | 0.737 |
| Obese | 82(81.2%) | 19(18.8%) | 2.0(1.1–3.8) | 0.033 | 2.3(1.2–4.5) | 0.013 |
| Low or normal | 246(89.8%) | 28(10.2%) | 1.0 | | 1.0 | |
| **Education** | | | | | | |
| Intermediate | 356(88.8%) | 45(11.2%) | 1.4(0.7–2.7) | 0.342 | 2.4(1.1–4.9) | 0.022 |
| University or above | 87(82.1%) | 19(17.9%) | 2.3(1.1–5.0) | 0.029 | 5.1(2.1–12.5) | <0.001 |
| Primary or below | 135(91.2%) | 13(8.8%) | 1.0 | | 1.0 | |
| **Employment status** | | | | | | |
| Others | 49(89.1%) | 6(10.9%) | 1.1(0.5–2.7) | 0.784 | 1.4(0.6–3.4) | 0.467 |
| Retired/homemakers | 123(84.2%) | 23(15.8%) | 1.6(0.9–2.7) | 0.087 | 2.4(1.2–4.6) | 0.010 |
| Working | 406(89.6%) | 47(10.4%) | 1.0 | | 1.0 | |
| **Gender** | | | | | | |
| Female | 279(86.4%) | 44(13.6%) | 1.5(0.9–2.4) | 0.124 | 1.3(0.8–2.3) | 0.289 |
| Male | 299(90.1%) | 33(9.9%) | 1.0 | | 1.0 | |

| | Barrier factors | | | | | |
|---|---|---|---|---|---|---|
| **Variable** | **High score** | **Very low score** | **Unadjusted** | | **Adjusted** | |
| | | | **OR (95% CI)** | **P-value** | **OR (95% CI)** | **P- value** |
| | External circumstances | | | | | |
| **BMI** | | | | | | |
| Overweight | 224(80.0%) | 56(20.0%) | 1.8(1.1–2.9) | 0.012 | 1.9(1.2–3.0) | 0.010 |
| Obese | 78(77.2%) | 23(22.8%) | 2.1(1.2–3.8) | 0.015 | 2.1(1.2–3.9) | 0.015 |
| Low or normal | 240(87.9%) | 33(12.1%) | 1.0 | | 1.0 | |
| **Education** | | | | | | |
| Primary or below | 120(81.1%) | 28(18.9%) | 1.3(0.8–2.1) | 0.338 | 1.3(0.7–2.2) | 0.391 |
| University or above | 84(79.2%) | 22(20.8%) | 1.5(0.8–2.5) | 0.176 | 1.6(0.9–2.8) | 0.098 |
| Intermediate | 339(84.5%) | 62(15.5%) | 1.0 | | 1.0 | |
| **Employment status** | | | | | | |
| Working | 380(83.9%) | 73(16.1%) | 1.1(0.6–1.8) | 0.825 | 1.2(0.7–2.1) | 0.581 |
| Others | 39(70.9%) | 16(29.1%) | 2.2(1.1–4.6) | 0.035 | 2.5(1.2–5.5) | 0.017 |
| Retired/homemakers | 124(84.4%) | 23(15.6%) | 1.0 | | 1.0 | |
| | Limited self-efficacy and competence | | | | | |
| **BMI** | | | | | | |
| Overweight | 246(87.9%) | 34(12.1%) | 1.4(0.8–2.4) | 0.266 | 1.7(0.9–3.1) | 0.077 |

(*Continued*)

**Table 3.** (Continued)

| | | | | | | |
|---|---|---|---|---|---|---|
| Obese | 81(80.2%) | 20(19.8%) | 2.4(1.3–4.6) | 0.007 | 2.8(1.4–5.5) | 0.003 |
| Low or normal | 249(90.9%) | 25(9.1%) | 1.0 | | 1.0 | |
| **Age group** | | | | | | |
| 40–59 | 231(85.2%) | 40(14.8%) | 1.8(1.0–3.2) | 0.044 | 1.8(1.0–3.3) | 0.050 |
| 60–75 | 140(88.6%) | 18(11.4%) | 1.3(0.7–2.5) | 0.471 | 1.3(0.5–3.0) | 0.564 |
| 20–39 | 206(91.2%) | 20(8.8%) | 1.0 | | 1.0 | |
| **Education** | | | | | | |
| Intermediate | 361(90.0%) | 40(10.0%) | 0.82(0.45–1.50) | 0.525 | 1.3(0.7–2.6) | 0.443 |
| University or above | 85(80.2%) | 21(19.8%) | 1.8(0.9–3.6) | 0.091 | 3.5(1.5–8.4) | 0.004 |
| Primary or below | 130(88.4%) | 17(11.6%) | 1.0 | | 1.0 | |
| **Employment status** | | | | | | |
| Others | 47(85.5%) | 8(14.5%) | 1.4(0.6–3.1) | 0.463 | 1.7(0.7–4.1) | 0.223 |
| Retired/homemakers | 124(84.9%) | 22(15.1%) | 1.5(0.9–2.6) | 0.129 | 1.5(0.7–3.1) | 0.316 |
| Working | 405(89.4%) | 48(10.6%) | 1.0 | | 1.0 | |
| **Gender** | | | | | | |
| Female | 270(83.9%) | 52(16.1%) | 2.3(1.4–3.8) | 0.001 | 2.6(1.5–4.5) | 0.001 |
| Male | 306(92.2%) | 26(7.8%) | 1.0 | | 1.0 | |
| **Lack of perceived susceptibility, benefits and intentions** | | | | | | |
| **BMI** | | | | | | |
| Overweight | 223(79.6%) | 57(20.4%) | 1.8(1.1–2.9) | 0.012 | 1.9(1.2–3.0) | 0.008 |
| Obese | 90(89.1%) | 11(10.9%) | 0.85(0.41–1.75) | 0.652 | 1.0(0.5–2.1) | 0.985 |
| Low or normal | 239(87.5%) | 34(12.5%) | 1.0 | | 1.0 | |
| **Age group** | | | | | | |
| 20–39 | 180(79.6%) | 46(20.4%) | 1.9(1.1–3.4) | 0.028 | 1.9(1.0–3.7) | 0.068 |
| 40–59 | 233(86.0%) | 38(14.0%) | 1.2(0.7–2.2) | 0.523 | 1.2(0.6–2.2) | 0.644 |
| 60–75 | 139(88.5%) | 18(11.5%) | 1.0 | | 1.0 | |
| **Education** | | | | | | |
| Primary or below | 127(86.4%) | 20(13.6%) | 1.0(0.6–1.7) | 0.979 | 1.3(0.7–2.3) | 0.461 |
| University or above | 80(75.5%) | 26(24.5%) | 2.1(1.2–3.5) | 0.007 | 1.9(1.1–3.3) | 0.016 |
| Intermediate | 346(86.3%) | 55(13.7%) | 1.0 | | 1.0 | |
| **Perceived lack of physical capability** | | | | | | |
| **BMI** | | | | | | |
| Overweight | 212(75.7%) | 68(24.3%) | 1.9(1.2–3.0) | 0.003 | 1.7(1.1–2.7) | 0.017 |
| Obese | 69(68.3%) | 32(31.7%) | 2.7(1.6–4.7) | <0.001 | 2.4(1.4–4.2) | 0.003 |
| Low or normal | 234(85.7%) | 39(14.3%) | 1.0 | | 1.0 | |
| **Age group** | | | | | | |
| 40–59 | 208(76.8%) | 63(23.2%) | 2.3(1.4–3.8) | 0.001 | 1.9(1.2–3.3) | 0.010 |
| 60–75 | 107(67.7%) | 51(32.3%) | 3.6(2.1–6.1) | <0.001 | 2.1(1.1–4.0) | 0.026 |
| 20–39 | 200(88.5%) | 26(11.5%) | 1.0 | | 1.0 | |
| **Education** | | | | | | |
| Primary or below | 102(69.4%) | 45(30.6%) | 2.0(1.3–3.1) | 0.002 | 1.0(0.6–1.7) | 0.908 |
| University or above | 85(80.2%) | 21(19.8%) | 1.1(0.7–1.9) | 0.680 | 1.4(0.8–2.5) | 0.246 |
| Intermediate | 328(81.8%) | 73(18.2%) | 1.0 | | 1.0 | |
| **Employment status** | | | | | | |
| Working | 377(83.2%) | 76(16.8%) | 1.4(0.6–3.2) | 0.445 | 1.4(0.6–3.3) | 0.434 |
| Retired/homemakers | 90(61.6%) | 56(38.4%) | 4.3(1.8–10.0) | 0.001 | 3.4(1.4–8.2) | 0.008 |
| Others | 48(87.3%) | 7(12.7%) | 1.0 | | 1.0 | |

Note: OR = 1.0 is the reference category.

benefits and intentions", those with university level education or above were significantly more likely to be high in this barrier factor as compared to intermediate level education (adjusted OR 1.9, 95% CI 1.1–3.3, p = 0.016). The barrier factor of "perceived lack of physical capability" was significantly higher in those aged 40 to 59 and 60 to 75 compared to those aged 20 to 39 (adjusted OR 1.9, 95% CI 1.2–3.3, p = 0.010 and adjusted OR 2.1, 95% CI 1.1–4.0, p = 0.026, respectively) (Table 3).

### The smokers

**Total scores of motivators and barriers in smokers.** Those aged 60–75 were significantly more likely to have high barriers as compared to those aged 20–39 (adjusted OR 9.1, 95% CI 2.6–32.3, p = 0.001). Those with intermediate level education and university level education or above were significantly more likely to have high barriers as compared to those with primary level education or below (adjusted OR 5.0, 95% CI 1.2–20.6, p = 0.027 and adjusted OR 7.4, 95% CI 1.3–43.3, p = 0.025, respectively) (Table 4).

**Predictors of motivator factors in smokers.** For the factor of "significant others (family or friends)", primary level education or below and university level education or above were significantly associated with being low in this factor as compared to intermediate level education (adjusted OR 6.4, 95% CI 1.5–28.2, p = 0.014 and adjusted OR 9.6, 95% CI 2.0–47.1, p = 0.005, respectively) (Table 4).

**Predictors of barrier factors in smokers.** For the barrier factor of "limited self-efficacy and competence", smokers aged 40 to 59 and 60 to 75 were significantly more likely to be high in this factor compared to those aged 20 to 39 (adjusted OR 5.0, 95% CI 1.4–17.6, p = 0.012 and adjusted OR 8.0, 95% CI 1.9–33.4, p = 0.004, respectively). Intermediate education or above was significantly associated with being high in this barrier factor as compared to primary level education or below (adjusted OR 51.7, 95% CI 1.4–1850.9, p = 0.031). For the barrier factor of "lack of perceived susceptibility, benefits and intentions", those aged 60 to 75 were significantly more likely to be high in this factor compared to those aged 20 to 39 (adjusted OR 4.6, 95% CI 1.5–13.8, p = 0.007). Those with intermediate level education were significantly more likely to be high in this barrier factor as compared to primary level education or below (adjusted OR 4.4, 95% CI 1.3–14.6, p = 0.016). The barrier factor of "perceived lack of physical capability" was significantly higher in those aged 40 to 59 and those aged 60 to 75 as compared to those aged 20 to 39 (adjusted OR 3.6, 95% CI 1.2–10.7, p = 0.022 and adjusted OR 4.0, 95% CI 1.1–14.4, p = 0.034, respectively) (Table 4).

### The obese

**Total scores of motivators and barriers in the obese.** Among the obese, those aged 60–75 were significantly more likely to have low motivations as compared to those aged 40–59 (adjusted OR 4.4, 95% CI 1.1–17.2, p = 0.035, Table 5). Those with primary level education or below were significantly more likely to have low motivations as compared to those with intermediate level education or below (adjusted OR 4.0, 95% CI 1.2–14.1, p = 0.028). There was no significant association between sociodemographic variables and barriers (Table 5).

**Predictors of motivator factors in the obese.** For "external cues", those aged 60 to 75 as compared to those aged 40 to 59 were significantly more likely to be low in this motivation factor (unadjusted OR 4.1, 95% CI 1.4–11.4, p = 0.008). For "significant others (family or friends)", those with university level education or above were significantly more likely to be low in this factor as compared to intermediate level education (adjusted OR 35.0, 95% CI 2.1–587.2, p = 0.014) (Table 5).

**Table 4. Motivators and barriers among smokers: Overall score and factors.**

| Motivators (total score) | | | | | | |
|---|---|---|---|---|---|---|
| Variable | Motivated | Very Unmotivated | Unadjusted | | Adjusted | |
| | | | OR (95% CI) | P-value | OR (95% CI) | P- value |
| **BMI** | | | | | | |
| Low or normal | 75(75.0%) | 25(25.0%) | 1.8(0.9–3.6) | 0.097 | 2.2(1.0–4.5) | 0.040 |
| Obese | 26(78.8%) | 7(21.2%) | 1.4(0.5–3.7) | 0.527 | 1.0(0.4–2.9) | 0.928 |
| Overweight | 92(84.4%) | 17(15.6%) | 1.0 | | 1.0 | |
| **Physical activity levels** | | | | | | |
| Didn't meet recommendation | 122(71.8%) | 48(28.2%) | 44.6(3.6–551.7) | 0.003 | 50.4(4.0–628.7) | 0.002 |
| Meet recommendation | 71(98.6%) | 1(1.4%) | 1.0 | | 1.0 | |
| **Motivator factors** | | | | | | |
| Variable | High score | Very low score | Unadjusted | | Adjusted | |
| | | | OR (95% CI) | P-value | OR (95% CI) | P- value |
| **Outcome expectations** | | | | | | |
| **Employment status** | | | | | | |
| Working | 117(65.7%) | 61(34.3%) | 2.4(1.0–5.7) | 0.041 | 2.4(1.0–5.7) | 0.051 |
| Others | 17(77.3%) | 5(22.7%) | 1.4(0.4–5.1) | 0.570 | 1.3(0.4–4.9) | 0.653 |
| Retired/homemakers | 34(82.9%) | 7(17.1%) | 1.0 | | 1.0 | |
| **Physical activity levels** | | | | | | |
| Didn't meet recommendation | 106(62.4%) | 64(37.6%) | 3.6(1.7–7.5) | 0.001 | 3.6(1.7–7.5) | 0.001 |
| Meet recommendation | 61(85.9%) | 10(14.1%) | 1.0 | | 1.0 | |
| **External cues** | | | | | | |
| **Physical activity levels** | | | | | | |
| Didn't meet recommendation | 115(67.6%) | 55(32.4%) | 4.1(1.8–9.4) | 0.001 | NA | NA |
| Meet recommendation | 64(90.1%) | 7(9.9%) | 1.0 | | | |
| **Significant others (family or friends)** | | | | | | |
| **BMI** | | | | | | |
| Low or normal | 91(91.0%) | 9(9.0%) | 10.2(0.3–373.5) | 0.206 | 12.3(0.3–463.1) | 0.175 |
| Overweight | 106(97.2%) | 3(2.8%) | 2.9(0.1–118.7) | 0.578 | 2.6(0.1–111.3) | 0.613 |
| Obese | 33(100.0%) | 0(0.0%) | 1.0 | | 1.0 | |
| **Education** | | | | | | |
| Primary or below | 45(90.0%) | 5(10.0%) | 5.3(1.2–22.5) | 0.024 | 6.4(1.5–28.2) | 0.014 |
| University or above | 25(86.2%) | 4(13.8%) | 7.7(1.7–35.5) | 0.009 | 9.6(2.0–47.1) | 0.005 |
| Intermediate | 158(98.1%) | 3(1.9%) | 1.0 | | 1.0 | |
| **Barriers (total score)** | | | | | | |
| Variable | Low | Very high | Unadjusted | | Adjusted | |
| | | | OR (95% CI) | P-value | OR (95% CI) | P- value |
| **BMI** | | | | | | |
| Low or normal | 84(84.0%) | 16(16.0%) | 3.3(1.2–9.0) | 0.018 | 3.7(1.3–10.7) | 0.015 |
| Obese | 29(87.9%) | 4(12.1%) | 2.8(0.8–10.2) | 0.125 | 3.1(0.8–12.3) | 0.108 |
| Overweight | 103(94.5%) | 6(5.5%) | 1.0 | | 1.0 | |
| **Age group** | | | | | | |
| 40–59 | 91(89.2%) | 11(10.8%) | 2.3(0.7–7.3) | 0.173 | 2.6(0.8–8.7) | 0.114 |
| 60–75 | 44(80.0%) | 11(20.0%) | 4.8(1.5–15.9) | 0.009 | 9.1(2.6–32.3) | 0.001 |
| 20–39 | 80(95.2%) | 4(4.8%) | 1.0 | | 1.0 | |
| **Education** | | | | | | |
| Intermediate | 142(88.2%) | 19(11.8%) | 2.5(0.7–9.6) | 0.178 | 5.0(1.2–20.6) | 0.027 |
| University or above | 25(86.2%) | 4(13.8%) | 2.9(0.6–14.8) | 0.209 | 7.4(1.3–43.3) | 0.025 |

(*Continued*)

**Table 4.** (*Continued*)

| Variable | High score | Very low score | Unadjusted | | Adjusted | |
|---|---|---|---|---|---|---|
| | | | OR (95% CI) | P-value | OR (95% CI) | P- value |
| Primary or below | 48(94.1%) | 3(5.9%) | 1.0 | | 1.0 | |
| **Barrier factors** | | | | | | |
| **External circumstances** | | | | | | |
| **Age group** | | | | | | |
| 40–59 | 85(83.3%) | 17(16.7%) | 2.2(0.9–5.5) | 0.105 | 2.0(0.8–5.1) | 0.148 |
| 60–75 | 44(80.0%) | 11(20.0%) | 2.7(1.0–7.6) | 0.051 | 2.5(0.9–7.0) | 0.078 |
| 20–39 | 77(91.7%) | 7(8.3%) | 1.0 | | 1.0 | |
| **Physical activity levels** | | | | | | |
| Didn't meet recommendation | 140(82.4%) | 30(17.6%) | 2.9(1.1–8) | 0.036 | 2.7(1.0–7.4) | 0.053 |
| Meet recommendation | 66(93.0%) | 5(7.0%) | 1.0 | | 1.0 | |
| **Limited self-efficacy and competence** | | | | | | |
| **BMI** | | | | | | |
| Low or normal | 85(85.9%) | 14(14.1%) | 2.5(1.0–6.7) | 0.057 | 2.7(1.0–7.5) | 0.058 |
| Obese | 29(87.9%) | 4(12.1%) | 2.4(0.7–8.4) | 0.182 | 1.8(0.5–7.0) | 0.391 |
| Overweight | 102(93.6%) | 7(6.4%) | 1.0 | | 1.0 | |
| **Age group** | | | | | | |
| 40–59 | 87(85.3%) | 15(14.7%) | 4.0(1.2–13.6) | 0.024 | 5.1(1.5–17.7) | 0.010 |
| 60–75 | 48(87.3%) | 7(12.7%) | 3.5(0.9–13.3) | 0.067 | 7.1(1.7–28.8) | 0.006 |
| 20–39 | 81(96.4%) | 3(3.6%) | 1.0 | | 1.0 | |
| **Education** | | | | | | |
| Intermediate or above | 166(86.9%) | 25(13.1%) | 24.4(0.7–852.1) | 0.078 | 51.7(1.4–1850.9) | 0.031 |
| Primary or below | 50(100.0%) | 0(0.0%) | 1.0 | | 1.0 | |
| **Physical activity levels** | | | | | | |
| Didn't meet recommendation | 147(86.5%) | 23(13.5%) | 3.6(1.0–12.7) | 0.045 | 4.5(1.2–16.7) | 0.025 |
| Meet recommendation | 68(95.8%) | 3(4.2%) | 1.0 | | 1.0 | |
| **Lack of perceived susceptibility, benefits and intentions** | | | | | | |
| **Age group** | | | | | | |
| 40–59 | 89(87.3%) | 13(12.7%) | 1.7(0.6–4.6) | 0.288 | 2.1(0.7–5.6) | 0.163 |
| 60–75 | 44(80.0%) | 11(20.0%) | 3.0(1.1–8.5) | 0.034 | 4.6(1.5–13.8) | 0.007 |
| 20–39 | 78(91.8%) | 7(8.2%) | 1.0 | | 1.0 | |
| **Education** | | | | | | |
| Intermediate | 137(85.1%) | 24(14.9%) | 2.2(0.7–6.7) | 0.178 | 4.4(1.3–14.6) | 0.016 |
| University or above | 27(90.0%) | 3(10.0%) | 1.2(0.2–6.1) | 0.845 | 2.3(0.4–12.6) | 0.352 |
| Primary or below | 47(92.2%) | 4(7.8%) | 1.0 | | 1.0 | |
| **Physical activity levels** | | | | | | |
| Didn't meet recommendation | 143(84.1%) | 27(15.9%) | 3.0(1.0–8.9) | 0.043 | 3.3(1.1–9.9) | 0.033 |
| Meet recommendation | 67(94.4%) | 4(5.6%) | 1.0 | | 1.0 | |
| **Perceived lack of physical capability** | | | | | | |
| **Age group** | | | | | | |
| 40–59 | 82(81.2%) | 19(18.8%) | 4.2(1.4–12.3) | 0.009 | 3.6(1.2–10.7) | 0.022 |
| 60–75 | 41(74.5%) | 14(25.5%) | 6.2(2.0–19.3) | 0.001 | 4.0(1.1–14.4) | 0.034 |
| 20–39 | 80(95.2%) | 4(4.8%) | 1.0 | | 1.0 | |
| **Employment status** | | | | | | |
| Working | 154(86.5%) | 24(13.5%) | 1.3(0.3–5.6) | 0.703 | 1.5(0.3–6.7) | 0.604 |
| Retired/homemakers | 30(71.4%) | 12(28.6%) | 3.4(0.7–15.7) | 0.117 | 2.9(0.6–14.9) | 0.192 |
| Others | 19(90.5%) | 2(9.5%) | 1.0 | | 1.0 | |

(*Continued*)

**Table 4.** (Continued)

| Physical activity levels | | | | | | |
|---|---|---|---|---|---|---|
| Didn't meet recommendation | 135(79.4%) | 35(20.6%) | 5.5(1.7–17.8) | 0.005 | 5.6(1.7–18.7) | 0.005 |
| Meet recommendation | 68(95.8%) | 3(4.2%) | 1.0 | | 1.0 | |

Note: OR = 1.0 is the reference category.

**Predictors of barrier factors in the obese.** Those with university level education or above were significantly more likely to be high in the barrier factor of "lack of perceived susceptibility, benefits and intentions" (adjusted OR 64.7, 95% CI 7.2–578.9, p<0.001) compared to those who were of primary level education or below. The retired or homemakers were significantly more likely to be high in the barrier factor of "limited self-efficacy and competence" (adjusted OR 4.9, 95% CI 1.2–20.0, p = 0.026) compared to those who were in workforce. Those of university level education or above were significantly more likely than those of intermediate level education to be high in the barrier factor of "perceived lack of physical capability" (adjusted OR 5.7, 95% CI 1.2–28.4, p = 0.033) (Table 5). Overall, race (Chinese, Malay or Indian) was not a significant factor determining the likelihood of having motivators or barriers to heart healthy behaviors.

## Discussion

This study examined the patterns of motivators and barriers influencing heart health behaviors in adult population segments with three behavior-modifiable risk factors (physical inactivity, smoking and obesity). Overall motivations were more likely to be low in those with lower education (for the physically inactive and the obese) whereas overall barriers were more likely to be high in those with higher education (for the physically inactive and the smokers). This finding of overall higher perceived barriers in the well-educated was also seen in Japan [12].

We found that individuals with lower education were more likely to have low motivation factor of "outcome expectations" (for the physically inactive). This indicates that people with lower education tended not to prioritize long-term health benefits of heart health behavior. This finding is in line with studies that people with lower socio-economic status in terms of income and education were less likely to prioritize long-term health benefits [24, 25]. Another finding is that individuals with lower education had low motivation factor of "external cues" which represent public health campaigns or advice from healthcare professionals (for the physically inactive). This affirms existing evidence of generally poor receptivity of health promotion efforts [26] among people of low education. However, in our study, people with lower education were found to be motivated by "significant others (family or friends)" (for the physically inactive and the obese). As with the studies that found the beneficial effects of social influence in improving uptake of physical activity, it may be useful to engage family and friends in interventions to increase motivation in the lower education group. Interventions for this group should also consider shifting individual's priorities to health benefits and guiding them to see the immediate relevance of health benefits to them.

This study showed that a key barrier to heart health behaviors in individuals with higher education is "limited self-efficacy and competence" (for the physically inactive and the obese). A possible explanation could be that the highly educated do place importance on the benefits of health behaviors, and hence more have tried to change, albeit lack of success. This finding is different from studies in Saudi Arabia [27] and Brazil [25]. Our findings also showed that the well-educated perceived had low perceived susceptibility to the risk of heart disease (for the

**Table 5. Motivators and barriers among the obese: Overall score and factors.**

| Variable | Motivated | Very Unmotivated | Unadjusted | | Adjusted | |
|---|---|---|---|---|---|---|
| | | | OR (95% CI) | P-value | OR (95% CI) | P- value |
| **Motivators (total score)** | | | | | | |
| **Age group** | | | | | | |
| 20–39 | 28(87.5%) | 4(12.5%) | 1.6(0.4–5.9) | 0.484 | 4.3(0.9–21.0) | 0.070 |
| 60–75 | 29(70.7%) | 12(29.3%) | 4.2(1.4–12.2) | 0.009 | 4.4(1.1–17.2) | 0.035 |
| 40–59 | 59(90.8%) | 6(9.2%) | 1.0 | | 1.0 | |
| **Education** | | | | | | |
| Primary or below | 35(72.9%) | 13(27.1%) | 3.8(1.4–10.0) | 0.008 | 4.0(1.2–14.1) | 0.028 |
| University or above | 6(75.0%) | 2(25.0%) | 3.3(0.6–19.9) | 0.185 | 6.7(0.9–51.9) | 0.068 |
| Intermediate | 76(90.5%) | 8(9.5%) | 1.0 | | 1.0 | |
| **Employment status** | | | | | | |
| Others | 10(76.9%) | 3(23.1%) | 2.6(0.6–10.9) | 0.195 | 2.2(0.5–10.3) | 0.324 |
| Retired/homemakers | 34(77.3%) | 10(22.7%) | 2.4(0.9–6.4) | 0.077 | 1.3(0.3–5.2) | 0.685 |
| Working | 72(88.9%) | 9(11.1%) | 1.0 | | 1.0 | |
| **Physical activity levels** | | | | | | |
| Low | 80(79.2%) | 21(20.8%) | 4.8(1.1–21.8) | 0.043 | 5.5(1.1–26.5) | 0.035 |
| Meet recommendation | 36(94.7%) | 2(5.3%) | 1.0 | | 1.0 | |

| **Motivator factors** | | | | | | |
|---|---|---|---|---|---|---|
| Variable | High score | Very low score | Unadjusted | | Adjusted | |
| | | | OR (95% CI) | P-value | OR (95% CI) | P- value |
| **Outcome expectations** | | | | | | |
| **Race** | | | | | | |
| Chinese | 53(66.3%) | 27(33.8%) | 1.9(0.8–4.5) | 0.170 | 1.8(0.7–4.6) | 0.197 |
| Indian | 13(72.2%) | 5(27.8%) | 1.4(0.4–5.0) | 0.591 | 1.3(0.4–4.9) | 0.672 |
| Malay | 31(77.5%) | 9(22.5%) | 1.0 | | 1.0 | |
| **Employment status** | | | | | | |
| Working | 54(65.9%) | 28(34.1%) | 1.9(0.8–4.5) | 0.140 | 1.7(0.7–4.2) | 0.219 |
| Others | 8(66.7%) | 4(33.3%) | 1.9(0.5–7.5) | 0.335 | 1.9(0.5–8.1) | 0.372 |
| Retired/homemakers | 35(79.5%) | 9(20.5%) | 1.0 | | 1.0 | |
| **Physical activity levels** | | | | | | |
| Didn't meet recommendation | 64(63.4%) | 37(36.6%) | 3.9(1.4–10.9) | 0.010 | 3.4(1.2–9.7) | 0.023 |
| Meet recommendation | 33(86.8%) | 5(13.2%) | 1.0 | | 1.0 | |
| **External cues** | | | | | | |
| **Age group** | | | | | | |
| 20–39 | 28(87.5%) | 4(12.5%) | 0.53(0.15–1.83) | 0.314 | 1.4(0.3–6.0) | 0.634 |
| 60–75 | 25(61.0%) | 16(39.0%) | 2.6(1.1–6.1) | 0.033 | 4.1(1.4–11.4) | 0.008 |
| 40–59 | 52(80.0%) | 13(20.0%) | 1.0 | | 1.0 | |
| **Education** | | | | | | |
| Primary or below | 31(64.6%) | 17(35.4%) | 3.2(1.4–7.6) | 0.007 | 3.2(1.1–9.0) | 0.028 |
| University or above | 3(37.5%) | 5(62.5%) | 8.6(1.8–40.7) | 0.007 | 13.8(2.5–77.8) | 0.003 |
| Intermediate | 71(85.5%) | 12(14.5%) | 1.0 | | 1.0 | |
| **Gender** | | | | | | |
| Male | 40(70.2%) | 17(29.8%) | 1.7(0.8–3.8) | 0.173 | 2.7(1.0–6.9) | 0.043 |
| Female | 65(80.2%) | 16(19.8%) | 1.0 | | 1.0 | |
| **Physical activity levels** | | | | | | |
| Didn't meet recommendation | 72(71.3%) | 29(28.7%) | 3.6(1.2–11.4) | 0.026 | 3.6(1.0–12.8) | 0.043 |
| Meet recommendation | 34(89.5%) | 4(10.5%) | 1.0 | | 1.0 | |

*(Continued)*

**Table 5.** (Continued)

| Significant others (family or friends) | | | | | | |
|---|---|---|---|---|---|---|
| **Age group** | | | | | | |
| 20–39 | 28(87.5%) | 4(12.5%) | 3.6(0.7–18.2) | 0.120 | 8.3(0.8–80.3) | 0.069 |
| 60–75 | 38(90.5%) | 4(9.5%) | 2.5(0.5–12.8) | 0.263 | 2.9(0.4–23.2) | 0.308 |
| 40–59 | 63(95.5%) | 3(4.5%) | 1.0 | | 1.0 | |
| **Education** | | | | | | |
| Primary or below | 43(89.6%) | 5(10.4%) | 2.0(0.5–8.0) | 0.327 | 3.2(0.4–23.8) | 0.246 |
| University or above | 6(75.0%) | 2(25.0%) | 6.5(1.0–43.1) | 0.054 | 35.0(2.1–587.2) | 0.014 |
| Intermediate | 79(95.2%) | 4(4.8%) | 1.0 | | 1.0 | |
| **Gender** | | | | | | |
| Female | 72(87.8%) | 10(12.2%) | 8.2(1.0–70.4) | 0.056 | 5.2(0.5–51.5) | 0.157 |
| Male | 56(98.2%) | 1(1.8%) | 1.0 | | 1.0 | |
| **Currently smoking** | | | | | | |
| No | 96(90.6%) | 10(9.4%) | 11.4(0.3–412.8) | 0.185 | 14.9(0.3–822.1) | 0.187 |
| Yes | 33(100.0%) | 0(0.0%) | 1.0 | | 1.0 | |
| **Barriers (total score)** | | | | | | |
| **Variable** | **Low** | **Very high** | **Unadjusted** | | **Adjusted** | |
| | | | OR (95% CI) | P-value | OR (95% CI) | P- value |
| **Age group** | | | | | | |
| 20–39 | 28(87.5%) | 4(12.5%) | 1.3(0.3–4.9) | 0.730 | 1.3(0.3–5.0) | 0.751 |
| 60–75 | 28(66.7%) | 14(33.3%) | 4.7(1.7–13.3) | 0.004 | 3.1(1–9.9.0) | 0.056 |
| 40–59 | 59(90.8%) | 6(9.2%) | 1.0 | | 1.0 | |
| **Employment status** | | | | | | |
| Others | 11(84.6%) | 2(15.4%) | 1.2(0.2–7.5) | 0.837 | 1.0(0.2–6.8) | 0.960 |
| Retired/homemakers | 30(68.2%) | 14(31.8%) | 4.2(1.6–10.9) | 0.003 | 2.2(0.7–7.0) | 0.167 |
| Working | 73(90.1%) | 8(9.9%) | 1.0 | | 1.0 | |
| **Gender** | | | | | | |
| Female | 64(79.0%) | 17(21.0%) | 2.0(0.8–5.4) | 0.147 | 1.6(0.6–4.6) | 0.387 |
| Male | 51(87.9%) | 7(12.1%) | 1.0 | | 1.0 | |
| **Barrier factors** | | | | | | |
| **Variable** | **High score** | **Very low score** | **Unadjusted** | | **Adjusted** | |
| | | | OR (95% CI) | P-value | OR (95% CI) | P- value |
| **External circumstances** | | | | | | |
| **Age group** | | | | | | |
| 40–59 | 54(83.1%) | 11(16.9%) | 1.5(0.4–5.3) | 0.510 | 1.4(0.4–5.1) | 0.569 |
| 60–75 | 28(66.7%) | 14(33.3%) | 3.7(1.1–12.9) | 0.039 | 2.4(0.6–9.6) | 0.208 |
| 20–39 | 28(87.5%) | 4(12.5%) | 1.0 | | 1.0 | |
| **Employment status** | | | | | | |
| Working | 69(85.2%) | 12(14.8%) | 1.7(0.2–11.6) | 0.613 | 1.8(0.3–13.1) | 0.552 |
| Retired/homemakers | 29(65.9%) | 15(34.1%) | 5.1(0.7–35.5) | 0.103 | 3.6(0.5–27.8) | 0.211 |
| Others | 12(92.3%) | 1(7.7%) | 1.0 | | 1.0 | |
| **Currently smoking** | | | | | | |
| No | 81(76.4%) | 25(23.6%) | 2.1(0.7–6.6) | 0.187 | 1.4(0.4–4.6) | 0.624 |
| Yes | 29(87.9%) | 4(12.1%) | 1.0 | | 1.0 | |
| **Limited self-efficacy and competence** | | | | | | |
| **Age group** | | | | | | |
| 40–59 | 55(84.6%) | 10(15.4%) | 2.9(0.6–14.7) | 0.200 | 2.9(0.4–19.6) | 0.276 |
| 60–75 | 29(70.7%) | 12(29.3%) | 6.8(1.4–34.5) | 0.020 | 6.7(0.9–50.7) | 0.065 |

(*Continued*)

**Table 5.** (Continued)

| | | | | | | |
|---|---|---|---|---|---|---|
| 20–39 | 30(93.8%) | 2(6.3%) | 1.0 | | 1.0 | |
| **Education** | | | | | | |
| Intermediate | 74(89.2%) | 9(10.8%) | 0.50(0.19–1.32) | 0.162 | 1.5(0.5–4.9) | 0.506 |
| University or above | 3(37.5%) | 5(62.5%) | 7.9(1.5–41.2) | 0.014 | 64.7(7.2–578.9) | <0.001 |
| Primary or below | 38(79.2%) | 10(20.8%) | 1.0 | | 1.0 | |
| **Employment status** | | | | | | |
| Others | 12(92.3%) | 1(7.7%) | 0.66(0.07–6.09) | 0.715 | 1.6(0.1–18.9) | 0.706 |
| Retired/homemakers | 30(66.7%) | 15(33.3%) | 4.2(1.6–10.8) | 0.003 | 4.9(1.2–20.0) | 0.026 |
| Working | 73(89.0%) | 9(11.0%) | 1.0 | | 1.0 | |
| **Gender** | | | | | | |
| Female | 61(75.3%) | 20(24.7%) | 4.7(1.5–14.7) | 0.009 | 5.0(1.3–19.6) | 0.022 |
| Male | 54(93.1%) | 4(6.9%) | 1.0 | | 1.0 | |
| **Perceived lack of physical capability** | | | | | | |
| **Age group** | | | | | | |
| 40–59 | 48(73.8%) | 17(26.2%) | 2.4(0.7–7.6) | 0.150 | 1.6(0.5–5.7) | 0.450 |
| 60–75 | 22(52.4%) | 20(47.6%) | 6.2(1.9–20.7) | 0.003 | 3.1(0.8–12.7) | 0.111 |
| 20–39 | 28(87.5%) | 4(12.5%) | 1.0 | | 1.0 | |
| **Education** | | | | | | |
| Primary or below | 27(56.3%) | 21(43.8%) | 3.2(1.4–6.9) | 0.004 | 1.9(0.8–4.7) | 0.163 |
| University or above | 4(50.0%) | 4(50.0%) | 4.2(0.9–18.9) | 0.063 | 5.7(1.2–28.4) | 0.033 |
| Intermediate | 67(80.7%) | 16(19.3%) | 1.0 | | 1.0 | |
| **Employment status** | | | | | | |
| Working | 64(78.0%) | 18(22.0%) | 2.7(0.4–18.2) | 0.318 | 2.7(0.4–19.8) | 0.316 |
| Retired/homemakers | 22(50.0%) | 22(50.0%) | 9.2(1.3–63.9) | 0.025 | 5.6(0.7–41.9) | 0.096 |
| Others | 12(92.3%) | 1(7.7%) | 1.0 | | 1.0 | |
| **Currently smoking** | | | | | | |
| No | 70(66.0%) | 36(34.0%) | 2.6(1.0–7.2) | 0.062 | 1.7(0.6–5.4) | 0.343 |
| Yes | 28(84.8%) | 5(15.2%) | 1.0 | | 1.0 | |

Note: OR = 1.0 is the reference category.

physically inactive) despite their poor heart health behaviors and did not anticipate the benefits of behavior change (for the obese). Strategies to improve self-efficacy and alter personal beliefs should consider developing role models and evidence-based information to allow the well-educated to better visualize the link between behavior modifiable risk factors and CVDs.

An interesting finding in the well-educated (among the obese, smokers and the physically inactive) was that they were less likely to have the motivation factor of "significant others (family or friends)" compared to those with intermediate levels of education. This finding is different from that of a study in Brazil where those with more schooling perceived more social support for behavioral change [25]. There are various reasons which may account for the differences. Firstly, unlike the present study, the Brazilian study did not specifically focus on population segments with poor health behaviors. In addition, various social and cultural contexts may explain the differences. Taken together, it becomes evident that groups with very low levels of education and groups with very high levels of education are less effectively motivated by significant others such as family and friends in comparison to those of intermediate level of education. This suggests that further research can focus on investigating the best ways to help the well-educated group gain social motivation, as social influence has been shown to have beneficial effects on health behaviors when applied positively [28, 29].

Lastly, an important finding was the oldest age group of 60 to 75 years was more likely to have overall low motivations (among the obese) and high barriers (among smokers) compared to other age groups. This finding indicates potential challenges in altering health behaviors for this age group. In terms of underlying motivator and barrier factors, the oldest age group in this study were more likely to be low in motivation factors of "external cues" (among the obese) and high in barrier factors of "lack of physical capability" (among the physically inactive and smokers), "limited self-efficacy and competence" and "lack of perceived susceptibility, benefits and intentions" (among the smokers). This is in contrast with a study in Japan that the elderly had overall better motivations compared to their younger counterparts [12]. These age-specific findings illuminate urgent intervention needs for the older adults with poor cardiovascular health behavior; as shown in this study, they may be particularly difficult to change as they have a myriad of barriers and lack motivators.

The strengths of this study included a large population-based sample of adult segments with behavior-modifiable risk factors, allowing for an understanding of the problems specifically affecting this group.

This study has a few limitations. The self-reported nature of the data means that findings may need to be treated with some caution due to recall or social desirability bias. Another limitation is related to the survey which limited options of motivators and barriers. It remains uncertain whether there would be more factors enabling or hindering heart healthy behaviors in this population.

In conclusion, this study found variability in the patterns of motivators and barriers affecting heart health behaviors in different population segments with behavior-modifiable risk factors. Those with lower education in general felt less motivated to make behavior changes for heart health while the well-educated were not fully convinced of the effectiveness of their actions in improving heart health. The well-educated also more keenly felt their past failures in behavioral change and lacked confidence in their ability to succeed. People aged 60 and above with poor behaviors were especially resistant to change and will likely need sustained efforts to change their attitudes. The patterns seen in the Asian population segments with behavioral risks will inform the design of future intervention and communication strategies addressing specific motivators and barriers.

## Supporting information

**S1 Data.**
(DTA)

## Acknowledgments

The authors would like to thank the administrative support from Vernon Kang and Linda Wee of the Singapore Heart Foundation.

## Author Contributions

**Conceptualization:** Zijuan Huang, Jien Sze Ho, Swee Yaw Tan, Woon Puay Koh, Terrance Siang Jin Chua, Sungwon Yoon.

**Data curation:** Zijuan Huang, Jien Sze Ho, Qai Ven Yap, Sungwon Yoon.

**Formal analysis:** Zijuan Huang, Jien Sze Ho, Qai Ven Yap, Yiong Huak Chan, Sungwon Yoon.

**Funding acquisition:** Lip Ping Low, Terrance Siang Jin Chua.

**Investigation:** Zijuan Huang, Jien Sze Ho, Sungwon Yoon.

**Methodology:** Zijuan Huang, Jien Sze Ho, Qai Ven Yap, Yiong Huak Chan, Swee Yaw Tan, Natalie Koh Si Ya, Lip Ping Low, Huay Cheem Tan, Woon Puay Koh, Sungwon Yoon.

**Project administration:** Zijuan Huang, Swee Yaw Tan, Sungwon Yoon.

**Resources:** Zijuan Huang, Huay Cheem Tan, Terrance Siang Jin Chua.

**Software:** Zijuan Huang, Qai Ven Yap.

**Supervision:** Zijuan Huang, Woon Puay Koh, Terrance Siang Jin Chua, Sungwon Yoon.

**Writing – original draft:** Zijuan Huang, Jien Sze Ho, Qai Ven Yap, Yiong Huak Chan, Sungwon Yoon.

**Writing – review & editing:** Zijuan Huang, Jien Sze Ho, Qai Ven Yap, Yiong Huak Chan, Swee Yaw Tan, Natalie Koh Si Ya, Lip Ping Low, Huay Cheem Tan, Woon Puay Koh, Terrance Siang Jin Chua, Sungwon Yoon.

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
