## [Decision Letter · Decision Letter 0]

13 Dec 2021

PONE-D-21-34711Patterns of motivators and barriers to heart health behaviors among adults with behavior-modifiable cardiovascular risk factors: a population-based survey in SingaporePLOS ONE

Dear Dr. Huang,

Thank you for submitting your manuscript to PLOS ONE. After careful consideration, we feel that it has merit but does not fully meet PLOS ONE’s publication criteria as it currently stands. Therefore, we invite you to submit a revised version of the manuscript that addresses the points raised during the review process.

We look forward to receiving your revised manuscript.

Kind regards,

Jagan Kumar Baskaradoss

Academic Editor

PLOS ONE

Journal Requirements:

Additional Editor Comments:

The manuscript is well written. However, reviewers have recommended minor revisions. We can consider the manuscript further once you have carried out some essential revisions suggested by our reviewers.

Reviewers' comments:

Reviewer's Responses to Questions

**Comments to the Author**

1. Is the manuscript technically sound, and do the data support the conclusions?

Reviewer #1: Yes

2. Has the statistical analysis been performed appropriately and rigorously? 

Reviewer #1: Yes

3. Have the authors made all data underlying the findings in their manuscript fully available?

Reviewer #1: Yes

4. Is the manuscript presented in an intelligible fashion and written in standard English?

Reviewer #1: Yes

5. Review Comments to the Author

Reviewer #1: This study aims to identify patterns of the motivators and barriers influencing heart health behaviors among multi-ethnic Asian adults with behavior-modifiable risk factors for heart disease, namely obesity, physical inactivity and smoking.

Overall this is a very well written paper and I have only minor comments for the authors:

1) Introduction: “identify motivators and barriers to uptake of heart health behavior ” – Meaning is unclear – rephrase

2) For the subheading “Behavior-modifiable Cardiovascular Risk Factors: Physical inactivity, Smoking and Obesity ” Add reference for each of the variables assessed (eg: MET, Smoking, BMI etc)

3) What was the distribution of the data for motivators are barriers – Why was mean score taken to dichotomize

4) Data was not provided as a supplementary file – please provide this

5) Nationality of respondents should be included to Table 1

6) Results: For- “outcome expectations”, those aged 60-75 were more likely to have low motivations as compared to those aged 40-59 (adjusted OR 1.6, 95% CI 1.0-2.5, p=0.048).” This is not statistically significant (OR included value 1). Revise the result accorindingly

6. PLOS authors have the option to publish the peer review history of their article (what does this mean?). If published, this will include your full peer review and any attached files.

Reviewer #1: No

---

## [Author Response · Author response to Decision Letter 0]

30 Dec 2021

Point-by-point response

1. Introduction: “identify motivators and barriers to uptake of heart health behavior ” – Meaning is unclear – rephrase

Response: Thanks for this comment. We have rephrased as “conducted a population survey to examine the factors that motivate and hinder uptake of heart health behavior in our Asian population. These motivators and barriers have previously been surveyed in the Western population (ref).”

2. For the subheading “Behavior-modifiable Cardiovascular Risk Factors: Physical inactivity, Smoking and Obesity” Add reference for each of the variables assessed (eg: MET, Smoking, BMI etc)

Response: References have been added for each of the variables assessed. 

3. What was the distribution of the data for motivators are barriers – Why was mean score taken to dichotomize

Response: The distribution of the data for barriers was normal and the distribution of the data for motivators was slightly skewed. Therefore, we feel that it would be reasonable to use mean-1sd as a cut off to dichotomise.

4. Data was not provided as a supplementary file – please provide this

Response: We have now provided the data as a supplementary file. 

5. Nationality of respondents should be included to Table 1

Response: All participants were Singaporeans or Permanent residents of Singapore as shown in the sample sub-section under Methods. In order to improve clarity, this has been stated again under Results in the sociodemographics sub-section.

6. Results: For- “outcome expectations”, those aged 60-75 were more likely to have low motivations as compared to those aged 40-59 (adjusted OR 1.6, 95% CI 1.0-2.5, p=0.048).” This is not statistically significant (OR included value 1). Revise the result accordingly.

Response: Thanks for this comment. We have amended the result to show 3 decimal places and to state that it is only borderline significant: “For “outcome expectations”, those aged 60-75 seemed more likely to have low motivations as compared to those aged 40-59, with borderline significance (adjusted OR 1.569, 95% CI 1.004-2.453, p=0.048).” We have also removed the outcome expectations from the discussion section given the marginally significant result.

---

## [Editor Report · Decision Letter 1]

5 Jan 2022

Patterns of motivators and barriers to heart health behaviors among adults with behavior-modifiable cardiovascular risk factors: a population-based survey in Singapore

PONE-D-21-34711R1

Dear Dr. Huang,

We’re pleased to inform you that your manuscript has been judged scientifically suitable for publication and will be formally accepted for publication once it meets all outstanding technical requirements.

Kind regards,

Jagan Kumar Baskaradoss

Academic Editor

PLOS ONE
---

## [Editor Report · Acceptance letter]

10 Jan 2022

PONE-D-21-34711R1 

Patterns of motivators and barriers to heart health behaviors among adults with behavior-modifiable cardiovascular risk factors: a population-based survey in Singapore 

Dear Dr. Huang:

I'm pleased to inform you that your manuscript has been deemed suitable for publication in PLOS ONE. Congratulations! Your manuscript is now with our production department. 

Kind regards, 

on behalf of

Dr. Jagan Kumar Baskaradoss 

Academic Editor

PLOS ONE